# Living Labs and Small and Medium-Sized Enterprises: A Symbioses Propelling Sustainable Innovation

**Julian Alexandrakis [1], Julia Hein [2] and Jan Kratzer [2,\*]**

1  Helmholtz Association, 10178 Berlin, Germany
2  Department of Technology and Management, Technische Universität Berlin, 10623 Berlin, Germany
\*  Correspondence: jan.kratzer@tu-berlin.de; Tel.: +49-162-436-1614

**Abstract:** Until recently, it had not been certain to what extent the integration of SMEs in Living Labs, as a special form of open innovation ecosystems, actually leads to successful commercialization of (sustainable) products and/or services. Today, while the effectiveness of publicly funded innovation projects is increasingly being debated, especially by European policymakers, who once hoped to solve the European paradox through this type of public-private-people partnership, this article can prove that in parts of the innovation process—and this includes, in particular, the *sustainable innovation output*—an increase in innovation performance on the site of the SMEs can be detected. Based on the concept of program evaluation, the effects on SMEs' innovation are determined with a preliminary qualitative study. The impact of Living Lab projects on innovation output, activities, capabilities, and knowledge flows of 12 European SMEs are then empirically tested in a quantitative study. Significant effects on the innovation performance of SMEs resulting from participation in Living Labs are identified. According to this study, Living Lab projects mainly influence the sustainable innovation output and knowledge flows of SMEs.

**Keywords:** Living Labs; sustainable innovation; SMEs

## 1. Introduction

Living Labs, which are increasingly spreading and publicly funded across Europe often in the context of transdisciplinary research and development, are a driving factor for future innovations. In addition, Living Labs support new and existing small and medium-sized enterprises (SMEs) in the innovation process by offering a space to co-create innovation with other innovation stakeholders. Nevertheless, the impact of Living Labs on the innovation performance of these SMEs is somewhat unexplored in the literature and lacks empirical evidence [1]. Living Labs are considered a rather new field of research that offers organizations new tools for creating innovation (Bergvall-Kåreborn et al., 2015). The expression is composed of two parts "Living" and "Lab". Thereby "Lab" describes intentional experimentation as it is carried out in a laboratory, while "Living" indicates that this is conducted in a real environment as opposed to an artificially created space. The literature on Living Labs focuses on the development of *Urban Living Labs* [2–4] and *Sustainable Living Labs* [5–8]. Since both concepts draw on the Living Lab approach, it is not possible to distinguish them clearly. Urban Living Labs differentiate themselves in that they have an explicit territorial focus on local, sustainable solutions. The problems that the Living Labs examine tend to be global, such as climate change and energy system transition [9]. In Sustainable Living Labs, the aspect of sustainability of products and services, as well as the reflection and change of routine-based lifestyles, is incorporated into the Living Lab approach [7].

Many Living Lab initiatives are single-project endeavors [10–12]. They distinguish themselves by being organized around a specific problem that leads to the development of a particular innovation. Living Labs, defined by a territorial area that serves as an arena

for several Living Labs focusing on a variety of challenges, can be called a Living Lab platform. The management of these area-defined Labs seeks to facilitate multiple Living Lab initiatives within a specific urban area and to foster favorable conditions [13]. The literature about the roles and functions of SMEs within Living Labs is rare. Hronszky and Kovács [14] identify two different roles that SMEs can have in Living Labs depending on the collaboration form. They might be involved on the user and/or on the manufacturer side as suppliers. On the supply or manufacturer side, they are responsible for contributing their expertise, skills, competencies, or even tangible resources for the development of the innovation. In contrast, on the user side, they are responsible for taking over the developed innovation, making use of it for their business, and in the case of a new product or service innovation, they might eventually offer it on the market. In turn, SMEs provide the Living Lab network with openness, knowledge, and data. They provide inputs for testing and stimulate discussions and creativity in such a way as to leave space for interactivity with the end-users [5,14]. At an early stage of the project, Living Labs provide SMEs with a practical approach to involving customers and users to gain feedback on prototypes in development [5,15,16]. Hence, Living Lab projects allow SMEs to test their products with potential users, receive feedback, and provide them with information early in the development phase to make necessary product adjustments or improvements. On the basis that the step of commercialization after the invention is essential for talking about innovation, and that SMEs are good at inventions but lack resources for commercialization, Living Labs can support them by testing market demand and customer value [17,18]. Santoro and Conte [19] are even going a step further and state that Living Labs can stimulate a user-driven market creating demand for new products or services early on in the development process. Lievens et al. [20] and Schaffers and Turkama [21] highlight the support Living Labs offer for SMEs to expand into new markets and achieve international turnover. Last but not least, Living Labs reduce the R&D costs of SMEs since the projects are often publicly subsidized, with SMEs paying only 20–25% of the project costs [15].

This study focuses on the explicit connection between Living Lab projects that are publicly funded and participating SMEs. Instead of investigating how the Living Lab projects influence a specific aspect or feature of a particular stakeholder, the innovation performance of SMEs as a result of participating in Living Labs and the vital and inherent interconnection as synergic systems are the core of the study. With this, the presented research effort contributes to research on interdisciplinarity and Living Labs as expression, as well as the sustainable economic and societal impact of transdisciplinary projects.

## 2. Measuring Innovation Performance of SMEs as a Result of Living Lab Participation

The potential effects of Living Lab projects on SMEs' innovation performance are divided according to the Oslo Manual into innovation output/outcome, activities, capabilities, and knowledge flows [22]. In this study, innovation output is defined as the result of the Living Lab project and includes all the activities SMEs undertake within the Living Lab project that might also have an impact on future innovation. Innovation activities include all the activities SMEs undertake within the Living Lab project that might also have an impact on future innovation. They include R&D, engineering, design, and other creative work, as well as the acquisition of tangible assets, and can create knowledge or information that is not necessarily used to develop the specific output of the Living Lab but can be used for the development of upcoming innovation. Innovation capabilities include design capabilities like engineering design, product design, and design thinking, as well as digital and technical expertise and competencies linked to the participation of the SME in the Living Lab project. Knowledge flows refer to the exchange and transmission of knowledge generated by the Living Lab project. It covers the increase in access to external sources of knowledge for innovating, innovation collaboration, and Open Innovation aspects of the Living Lab project.

The effects summarized in Table 1 are the result of thorough literature research structuring the benefits of Living Labs for SMEs according to criteria relevant for measuring

innovation performance. They will be discussed and adopted in interviews with Living Lab practitioners and SMEs, which then form the basis for the subsequent quantitative main study.

**Table 1.** Effects of Living Labs on the performance of SMEs.

| Dimension | Effects on SMEs' Innovation Performance | References |
|---|---|---|
| Innovation Output | Improved product | [15,23,24] |
| | Introducing new product or service | [25–27] |
| Innovation activities | Access to innovation services | [28] |
| | Acquisition of complementary resources and finance | [26–28] |
| | Reduced (R&D) costs | [15,29] |
| | Attracting private financing (for R&D) | [15] |
| | Test product-market fit | [5,15,18,19,25] |
| Innovation capabilities | Increased knowledge and competence level | [15,26,27,29] |
| | Access to new technologies | [29] |
| | Improved development/innovation process | [15,26–28] |
| | Reduced risks of product or process development | [15] |
| | Creation of Open Innovation culture/Develop an attitude for collaboration | [5,17,19,26,27] |
| Knowledge flows | New knowledge resources | [19,28] |
| | Access to new ideas | [29] |
| | Involving user and other external viewpoints in innovation process | [5,15,16] |
| | New market access | [20,21,28,29] |
| | More extensive social network | [26,27] |
| | Knowledge exchange and collaboration | [30] |

## 3. Research Design: Qualitative Research

This approach uses qualitative and quantitative data collection techniques and analysis procedures one after the other (sequential). It does not combine them, i.e., quantitative data are analyzed quantitatively, and qualitative data are analyzed qualitatively. A sequential exploratory research design is used. Program evaluation is best suited for mixed methods because complex and long-term projects often require researchers to connect several studies to reach the research objective. Mixed methods combine the advantages of qualitative and quantitative data collection and offset each other's weaknesses [31].

Only projects that demonstrate a Living Lab approach are included in the study, which means that they do not necessarily have to be labeled as such but, according to the Living Lab project website, display relevant characteristics. Furthermore, just Living Lab projects with a focus on sustainability in the urban context and the area of mobility and energy are considered in this study. The investigated projects are located within the EU and publicly supported through either regional, national, or EU budgets. All Living Lab projects have multiple SMEs participating in the project. Living Labs with a single SME as initiator are excluded. An overview of the criteria the Living Lab projects must fulfill is summarized in Table 2.

**Table 2.** Criteria for Living Lab projects and participating SMEs under investigation.

| Dimension | Criteria |
| --- | --- |
| Living Lab project | Living Lab approach applied<br>Focus on sustainability in an urban context<br>Area of mobility and energy<br>Within the EU<br>Publicly supported/funded<br>Ended or running<br>Multiple SMEs as partners |
| SME | Staff headcount < 250<br>Turnover $\leq$ 50 million Euro or balance sheet total $\leq$ 43 million Euro |

To identify applicable Living Lab projects, the Living Labs of the ENoLL network and *CORDIS*, the database of the European Commission with all EU-funded projects, have been filtered, scanned, and investigated for suitable projects. After identifying the projects, the participating companies were examined to clarify whether they fulfill the standards determining them as SMEs. This was accomplished by checking their online website for information. For German companies, the online platform for company data *Unternehmensregister* was scanned. After detecting the companies as SMEs, the Living Lab managers were contacted via e-mail or telephone and asked for contact details of the persons responsible for the participating SMEs and other projects they might know of that meet the criteria.

*3.1. Sampling, Semi Structured-Interview, and Analysis*

The survey instrument used for the preliminary study is an interview guide, on which the semi-structured interviews with Living Lab managers and SMEs are based. Five persons involved are selected according to availability and willingness to participate. This should ensure their ability to express their experiences and thoughts in an articulate and reflective manner. To recruit interview partners, invitations are sent to them via email, identified from the Living Lab projects' websites. In most cases, a second email is sent to them as a reminder. After that, attempts are made to contact them by phone. Based on similar precedent studies and the relatively small whole population, the sample for the preliminary research should include at least four interviews [32].

To analyze the interviews, the recorded audio files are first transcribed. Part of the interview is paraphrased, and part is transcribed verbatim [33]. Furthermore, a structuring technique is used to investigate the subject of the interviews related to the research questions [34]. Following this, the transcripts are coded, i.e., the data is labeled and systematized [35]. Therefore, different criteria must ensure the quality of the code, namely objectivity, reliability, and validity. The analysis of the interviews is based on the basic elements of qualitative content analysis, according to Mayring [34]. Table 3 lists the analyzed interviews. First, two Living Lab managers were interviewed, whose evaluations were then complemented by three companies that participated in Living Lab projects. The interview responses are analyzed in four phases according to the general procedure of content analysis [36], (a) Transcription, (b) Individual analysis, (c) Generalized analysis, and (d) Control.

**Table 3.** Overview of conducted interviews.

| No. | Synonym | Living Lab Project | Location | Role |
| --- | --- | --- | --- | --- |
| 1 | XM | Living Lab project X | Location X | Living Lab manager |
| 2 | FHM | Distribute | Berlin, Germany | Living Lab manager Technische Universität Berlin |
| 3 | RMC | Distribute | Berlin, Germany | Participating company insel-projekt.berlin UG |
| 4 | MKC | Adaptive City Mobility | Munich, Germany | Participating company Remoso GmbH |
| 5 | JAC | Distribute | Berlin, Germany | Participating company Adomeit Group GmbH |

*3.2. Results and Hypotheses*

Regarding the innovation output, the preliminary study has shown that both existing processes and products have been improved and that new services and products have been developed within the Living Labs. The product improvements are, therefore, incremental innovations in the product or process area. Three of the respondents described the Living Lab project leading to a new service or custom solution. It should be noted that two of the respondents, a Living Lab manager, and a company representative, took part in the same project. It is still unclear whether the service can be profitably replicated on the market in the future. The problem of profitable business models also shows that Living Lab projects are still strongly focused on the development of innovations and less on the creation of a sustainable business model. This is also highlighted by Alexandrakis and Hager [25], who emphasize the importance of integrated business model development at an early stage of a Living Lab. By analyzing and testing the appropriate business and market assumptions during the project, the acceptance and diffusion of the innovation by users are promoted. The fact that changes to the product are still necessary even after the Living Lab has been completed is also highlighted by the following statement of one of the respondents:

> "[...] our focus was this multi-mode. Of course, we have always adapted it a bit, developed it further and of course also transferred it, not into a standard solution, but a solution that is also used by our customers [...]. This is already used by us in customer projects, just not in this final stage of development." (MKC, 2020)

Nevertheless, it can be assumed from the preliminary study that the participation of SMEs in a Living Lab project has a positive influence on their innovation output, which is manifested in an improved or new product or service. The following hypotheses are derived from this:

**H1:** Living Lab projects have a positive effect on the innovation output of SMEs.

The benefits of Living Labs on innovation activities of SMEs were summarized. These activities cover a wide range and were often not empirically proven. Since participation in the Living Lab itself can already be counted as an innovation activity, the focus is increasingly on methods and resources that SMEs can also use in their innovation process in the future. The test of a product-market fit was emphasized as an advantage by all respondents. One example is stated below:

> "So for us, it is concretely positive that we could try out the solutions and concepts we have with the partners. [...] The feedback was valuable and to try it out in everyday life in the first place." (JAC, 2020)

Testing leads to product improvements based on customer feedback. It also results in higher product awareness and higher market acceptance which is underlined by the following statement:

> "So I do think that methodologically there were already approaches involved that were potentially beneficial for them [SMEs]." (FHM, 2020)

More than 50% of the interviewees mention that access to innovation services is advantageous for innovation activities. In this context, access to scientific institutions is mentioned positively. Most notable are methodical and organizational services, which were adopted and stimulated by the Living Lab project management. In contrast, less than 50% of the respondents mention the acquisition of complementary resources as beneficial for innovation activities. Mentioned resources include external development services, legal reports, and the scientific part of the project represented by other project partners.

The respondents focused less on financial advantages. The benefit of reduced costs and the acquisition of follow-up financing are only mentioned by one interviewee, respectively. The two respondents state that the Living Lab project led to follow-up subsidized projects as well as the co-financing of other developments outside the project. This is indicated by the following statement:

> "[. . .], because we also have developments in our company that are not only, no SME does, part of the project, but it must also correspond to the product philosophy, the product strategy and there we have of course also new interfaces, new operating concepts, etc. and also in the software, of course, adapted some things with these grants from this project of course." (MKC, 2020)

In summary, the preliminary study suggests that Living Labs also have a positive impact on the innovation activity of SMEs, leading to the following hypothesis.

**H2:** Living Lab projects have a positive effect on the innovation activities of SMEs.

The innovation capabilities include, above all, the increased knowledge and competence level resulting from the Living Lab project. This is also the most frequent answer by the respondents. All of them mentioned this as an advantage. The answers range from increased technology knowledge or competency to market knowledge and even the experience or expertise to develop a sustainable business model. In addition to technical or economic knowledge, methodological competence is also mentioned. These are highlighted by the following two respondents:

> "We always call it technology knowledge or technology skills. They just already existed. So that's where competence has already been built up." (XM, 2020)

> "..., but also beyond that, of course, kind of how you build business models." (FHM, 2020)

The second most common answer relates to the sustainable development of an Open Innovation culture and improved attitudes toward cooperation. Although two of the companies mentioned that there already is a very innovative culture within the company, at least one of them emphasizes that especially the discussions with users have led to a change in culture. On the other hand, in the following statement, one company speaks of a completely new dynamic within the company that was sparked by the project.

> "I think you've also really given a little bit of momentum to the company through this research project." (MKC, 2020)

Over 50% of the interviewees see the reduced risk in product development as very positive. This is mainly due to the subsidies for the project, the acquisition of follow-up projects, and the way in which product development is carried out. One respondent describes the reduced risk related to financial support as follows:

> "Now that probably has less to do with the vehicle and more to do with the grants that you get as an SME, and of course, you're a little more willing to take risks there, because of course I can digest that better." (MKC, 2020)

In contrast, not even half of the respondents name access to new technologies and the improved innovation/development process as advantages of Living Lab projects. Regarding access to new technologies, a programming course was an essential component

for *remoso GmbH*. A large consortium and the methodological experience of a co-creation process are decisive for an improved innovation process.

In summary, the following hypothesis is derived for innovative capabilities:

**H3:** Living Lab projects have a positive effect on the innovation capabilities of SMEs.

In the area of knowledge flows, all respondents confirmed that external viewpoints by users were included in the innovation process and were also perceived as valuable, highlighted by the following statement:

> "Basically, the format is very attractive to all stakeholders in terms of user engagement." (JAC, 2020)

Four out of five respondents said that the Living Lab project gave them access to new ideas and a broader network. Living Lab projects have both generated new ideas within the project and stimulated new ideas within the companies. Nevertheless, it is worth mentioning that a representative of an SME doubts that Living Lab projects are a source of ideas. Julius Adomeit was disappointed during the project that the users brought in only a few ideas or no ideas at all. On the other hand, the *Adomeit Group GmbH* was able to benefit from a broader network both within the project and beyond.

> "Yes, in principle, continue to expand the network, because we were, so to speak, from the field and also present with the other cargo bike in the relevant forums or events, but through the project, we have put out much more feelers in the direction of this service industry logistics." (JAC, 2020)

Furthermore, 60% of respondents indicate that there has been an exchange of knowledge and positive cooperation, which may also have an impact on the innovation performance of SMEs in the future. Participating companies are often still in contact after the project has ended. Almost half of the respondents mentioned access to new knowledge resources as beneficial. The scientific institutions played a major role in this, as they provided access to meetings outside the project, which would not have been possible otherwise. Michael Keun of the *remoso GmbH* states:

> "We had access to the ICT-EM III meetings in Berlin. Of course, that was quite fancy for us as a small SME." (MKC, 2020)

Only one respondent indicates that the project has also opened up new markets. With its project, *remoso GmbH* has won the Innovation Prize for SMEs several times, and the concept has aroused great interest in the Asian region, especially in India. Nevertheless, the most frequently mentioned characteristics are those within the area of knowledge flows. Therefore, the following hypothesis can be concluded from the preliminary qualitative study.

**H4:** Living Lab projects have a positive effect on knowledge flow in SMEs.

## 4. Research Design: Quantitative Study

The population for this study includes all SMEs that have participated in Living Lab projects within the EU, which have several SMEs as partners, are publicly funded, and have been active in the field of energy and mobility in urban areas. The ENoLL has 38 partners in the field of Mobility and Energy [37]. Assuming that each of them has carried out a project that meets the requirements of two SMEs, the assumed population is 72 companies. For gathering the data a standardized online survey is used. After the questionnaire has been completely programmed, the questions and the structure are checked for clarity, relevance of content, comprehensiveness, and accuracy [38]. For this purpose, the aspects mentioned above are checked with the help of a pretest. Here, the validity of the content is reviewed to determine whether the respondents understand the questions correctly. Further adjustments and improvements are made in the subsequent discussion and through Unipark's pretest comment function. In the same way, various

technical aspects of the functionality are checked, such as the progress bar and the screen output.

All participants are invited to the study via personal e-mail. The e-mail was sent to the participants in both English and German using the Unipark survey tool. Care was taken to send the e-mails at an advantageous time when a high response rate can be expected. Two reminders were then sent to the respondents at an interval of two and five days. Since the recruitment of participants had already proven to be extremely difficult in the qualitative survey, some SMEs are additionally contacted by telephone and asked to participate. The survey was held between August 11, 2020, and August 19, 2020. In total, 39 SMEs from 12 different Living Lab projects were contacted. The survey was finally completed by 12 persons.

In the questionnaire, the variables are operationalized as follows. The main variable in this study is the innovation performance of SMEs. To make the variable measurable, it is split into four dimensions: innovation output, innovation activities, innovation capabilities, and knowledge flows of SMEs (Adapted from OECD and Eurostat [22]). The operationalizations and sources are shown in Tables 4–7.

**Table 4.** Operationalization of innovation output.

|  | Indicator | 7-Point Likert Scale |
|---|---|---|
| InnOut1 | The project enabled us to improve an existing product/service/process. | (7) strongly disagree to (1) strongly agree |
| InnOut2 | The project enabled us to introduce a new product/service/process to the market. | (7) strongly disagree to (1) strongly agree |
| InnOut3 | Through the project, we plan to introduce a new product/service/process to the market. | (7) strongly disagree to (1) strongly agree |

**Table 5.** Operationalization of innovation activities.

|  | Indicator | 7-Point Likert Scale |
|---|---|---|
| InnAct1 | The project gave us access to new innovation services. | (7) strongly disagree to (1) strongly agree |
| InnAct2 | Through the project, we were able to acquire complementary resources and finances. | (7) strongly disagree to (1) strongly agree |
| InnAct3 | The project enabled us to reduce (R&D) costs. | (7) strongly disagree to (1) strongly agree |
| InnAct4 | The project enabled us to attract private financing (for R&D). | (7) strongly disagree to (1) strongly agree |
| InnAct5 | Through the project, we were able to test the product-market fit. | (7) strongly disagree to (1) strongly agree |
| InnAct6 | We will continue to benefit from the innovation activities and use them for future developments/innovations. | (7) strongly disagree to (1) strongly agree |

**Table 6.** Operationalization of innovation capabilities.

|  | **Indicator** | **7-Point Likert Scale** |
|---|---|---|
| InnCap1 | Through the project, we have achieved a higher level of knowledge and competence. | (7) strongly disagree to (1) strongly agree |
| InnCap2 | The project gave us access to new technologies. | (7) strongly disagree to (1) strongly agree |
| InnCap3 | Through the project, we were able to improve our development/innovation process. | (7) strongly disagree to (1) strongly agree |
| InnCap4 | The project helped us to reduce the risk of our product/service/process development. | (7) strongly disagree to (1) strongly agree |
| InnCap5 | Through the project, we have created an Open Innovation culture. | (7) strongly disagree to (1) strongly agree |
| InnCap6 | The project improved our attitude towards collaboration. | (7) strongly disagree to (1) strongly agree |
| InnCap7 | We will continue to benefit from our acquired innovation capabilities and use them for future developments/innovations | (7) strongly disagree to (1) strongly agree |

**Table 7.** Operationalization of knowledge flows.

|  | **Indicator** | **7-Point Likert Scale** |
|---|---|---|
| KnoFlo1 | Through the project, we identified new knowledge resources. | (7) strongly disagree to (1) strongly agree |
| KnoFlo2 | The project gave us access to new ideas. | (7) strongly disagree to (1) strongly agree |
| KnoFlo3 | Through the project, we were able to involve users and other external viewpoints in the innovation process. | (7) strongly disagree to (1) strongly agree |
| KnoFlo4 | The project gave us access to new markets. | (7) strongly disagree to (1) strongly agree |
| KnoFlo5 | Through the project, we were able to expand our network. | (7) strongly disagree to (1) strongly agree |
| KnoFlo6 | The project has enabled us to exchange knowledge and collaborate. | (7) strongly disagree to (1) strongly agree |
| KnoFlo7 | We will continue to benefit from these knowledge flows and use them for future developments/innovations. | (7) strongly disagree to (1) strongly agree |

## 5. Results of Quantitative Study

To answer the main research question on the impact of Living Lab projects on the innovation performance of SMEs, a descriptive evaluation is chosen due to the small sample size of SMEs. Therefore, all dimensions are first analyzed separately. Table 8 shows the frequency distribution of the effects on the innovation output. The frequencies and percentages are indicated for the improvement of an existing product/service/process, the introduction of a new product/service/process, and the planned introduction of a new product/service/process.

**Table 8.** Frequency distribution of innovation output (*n* = 12).

| | Improvement of Existing Product/Service/Process | | Introduction of New Product/Service/Process | | Planned Introduction of New Product/Service/Process | |
|---|---|---|---|---|---|---|
| | Frequency | Percent | Frequency | Percent | Frequency | Percent |
| 1-Strongly disagree | 1 | 8.3 | 0 | 0 | 0 | 0 |
| 2-Disagree | 2 | 16.7 | 2 | 16.7 | 2 | 16.7 |
| 3-Slightly disagree | 1 | 8.3 | 1 | 8.3 | 0 | 0 |
| 4-Neutral | 2 | 16.7 | 5 | 41.7 | 2 | 16.7 |
| 5-Slightly agree | 1 | 8.3 | 1 | 8.3 | 5 | 41.7 |
| 6-Agree | 3 | 25 | 2 | 16.7 | 3 | 25 |
| 7-Strongly agree | 2 | 16.7 | 1 | 8.3 | 0 | 0 |
| Median | 4.5 | | 4.00 | | 5.00 | |
| Standard deviation | 2.065 | | 2.386 | | 1379 | |
| Top-2-boxes | 41.7% | | 25% | | 25% | |

The median indicates the middle score in the distribution or the score that divides the distribution in half. In this case, the median is 4.5 for improving an existing product/service/process, 4 for the introduction of a new product/service/process, and 5 for the planned introduction of a new product/service/process. Hence it cannot be assumed that the Living Lab project has had a positive impact on the innovation output of SMEs.

Furthermore, the top-2-boxes can be evaluated to get an impression of the respondents' support for Living Lab projects. For this, the two answer options "strongly agree" and "agree" are combined. Here it can be observed that at least 41% of the respondents state that the Living Lab project has led to a product/service/process improvement. Only 25% say that the project has led to the introduction of a new product/service/process, or that they plan to introduce it. Thus, it could be deduced that the Living Lab projects are more likely to generate incremental innovation. The fact that 25% of respondents still plan the innovation's introduction to the market might be because the SMEs do not have a business model for the new product or service after the project has ended. Therefore, Living Labs should focus on developing sustainable business models during the project.

These ambiguous results may be because respondents who indicated that the project enabled them to introduce a new product were not focused on improving an existing product. In other words, the survey results reflect the different focuses of the projects. Therefore, each data set must be reviewed individually to determine how many respondents indicated that the Living Lab project had an impact on one of the indicators. In this case, 7 out of 12 respondents either agree or strongly agree that the Living Lab project they participated in had an impact on one of these indicators. This means that 58.3% of respondents say that the Living Lab project has had a positive impact on their innovation output.

It turned out that the operationalization of this dimension would have achieved more precise results if there had been a question that asked whether a project has had an impact on either the improvement, new development, or planning of a product/service/process or whether it has had no impact at all. This way, multiple selections can be excluded. Besides, it would then be possible to weigh the attributes differently and include them in evaluating the innovation output.

Twenty-five percent of respondents agreed to the impact of Living Labs on access to innovation, which indicates that Living Lab projects have very little impact on access to innovation services or are not perceived as such by SMEs. A stronger focus should be placed on this during the implementation of Living Labs to strengthen SMEs' innovation management in the long run. Fifty percent of the respondents agree that the projects have enabled them to acquire new complementary resources. Living Lab projects seem to have less impact on financial benefits. While 33.3% of respondents agreed that the project led to reduced (R&D) costs, only 16.7% could obtain follow-up financing through

the project. The high standard deviation again indicates the diverse settings of Living Lab projects. This allows for the conclusion that the project funding does not directly lead to reduced development costs and that the budget should be used in a more targeted manner if necessary. This is of particular importance for policymakers. Five out of twelve respondents said that the project allowed them to test the product-market fit to examine the product's acceptance for a future market launch. However, this does not indicate a strong positive influence on the Living Lab projects. In conclusion, the respondents were asked to indicate whether they will continue to benefit from and use these innovation activities in the future. More than half of the respondents (58.3%) agreed with this. An overview of the analysis results can be found in Table 9.

**Table 9.** Descriptive analysis of innovation activities.

|  | Access to New Innovation Services | Acquire Complementary Resources | Reduce (R&D) Costs | Attract Private Financing (for R&D) | Test Product-Market Fit | Benefit and Use in the Future |
|---|---|---|---|---|---|---|
| Median | 4.5 | 5.5 | 4.5 | 2 | 5 | 6 |
| Standard deviation | 1.564 | 1.467 | 1.676 | 1.946 | 1.165 | 0.965 |
| Top-2-boxes | 25% | 50% | 33.3% | 16.7% | 41.7% | 58.3% |

The percentage of respondents who stated that they had acquired a higher level of knowledge and competence through the Living Lab Project was 66.7%, as indicated in Table 10. This high level of agreement is also reflected in the the median of 6. The standard deviation is also comparatively low at 0.793. The situation is different when it comes to access to new technologies. Only 16.7% say that they have discovered new technologies through the Living Lab. The median is around 4 and indicates that the Living Lab project has no impact on new technologies. Living Lab managers should put an extra focus on this if necessary and provide access to the latest technology through scientific institutions and other partners. Concerning the development process, 25% of respondents state that they have improved their process and reduced development risk through the Living Lab project. The median for the improvement of the innovation process is 5. Nevertheless, in the future, Living Lab managers should ensure that they give SMEs practical tips on how to use the individual methods from the projects for their own innovation process. Only 8.3% indicated that the Living Lab project had promoted a more open innovation culture. This may be due to the fact that openness to the input of external opinions and ideas is a prerequisite for participation in the Living Lab project, i.e., participating SMEs show this characteristic even before the project. The percentage of respondents who indicate that their attitude towards collaboration has improved as a result of the Living Lab project is 41.7%. The median is 5. This indicates that SMEs are more open to external opinions and ideas because they participated in a Living Lab project. Nevertheless, the standard deviation is relatively high, and there is a high range of responses. Overall, no trend can be derived from the survey results. Half of the respondents say that they will benefit from the generated innovation capabilities in the future. The comparatively low standard deviation shows that the respondents have a relatively common opinion.

**Table 10.** Descriptive analysis of innovation capabilities.

|  | Higher Level of Knowledge and Competence | Access to New Technologies | Improved Development/ Innovation Process | Reduced Risk of Development | Create Open Innovation Culture | Improved Attitude towards Collaboration | Benefit and Use in the Future |
|---|---|---|---|---|---|---|---|
| Median | 6 | 4 | 5 | 4.5 | 4.5 | 5 | 5.5 |
| Standard deviation | 0.793 | 1.371 | 0.793 | 1.505 | 1.138 | 1.528 | 0.866 |
| Top-2-boxes | 66.7% | 16.7% | 25% | 25% | 8.3% | 41.7% | 50% |

Only 25% of respondents indicated that they could identify new knowledge resources through the Living Lab project (Table 11). This is surprising, considering that Living Lab projects often involve collaboration with universities or other scientific institutions. In the future, a greater focus should be placed on this during the implementation of Living Lab projects to demonstrate this knowledge access to the participating SMEs so that they can make use of it even after the project has ended. The respondents expressed significantly higher approval of access to new ideas. The percentage of respondents who thought that the Living Lab project positively affected access to new ideas was 58.3%. The median of 6 also confirms this approval. Just under half of those surveyed said that the Living Lab project had enabled them to integrate users and external views into the innovation process. The percentage of respondents who have gained access to new markets through the Living Lab project is 41.7%. More than half of the respondents say that their network has expanded due to the Living Lab project. The median is 6, which is a clear sign of agreement. The standard deviation is also only slightly above 1. The Living Lab projects' effects are most evident in the exchange of knowledge and cooperation. The percentage of those surveyed who assessed this positively is 66.7%. Here, too, the respondents were asked whether they would benefit in the future from the knowledge flows gained through the Living Lab project and whether they would use them.

**Table 11.** Descriptive analysis of knowledge flows.

| | New Knowledge Resources | Access to New Ideas | Involve Users/ External Viewpoints in Innovation Process | Access to New Markets | Expand Network | Exchange Knowledge and Collaborate | Benefit and Use in the Future |
|---|---|---|---|---|---|---|---|
| Median | 4.5 | 6 | 5 | 4 | 6 | 6 | 6 |
| Standard deviation | 1.730 | 1.215 | 1.586 | 1.784 | 1.267 | 1.044 | 0.793 |
| Top-2-boxes | 25% | 58.3% | 41.7% | 41.7% | 58.3% | 66.7% | 66.7% |

## 6. Discussion and Conclusions

When looking at the quantitative results, it becomes clear that Living Lab projects significantly influence the innovation output of SMEs and their knowledge flows. SMEs indicate that they will benefit in the future from knowledge flows and emphasize the positive effect of access to new ideas. In addition, the expansion of their network and exchange of knowledge and collaboration are the indicators with the highest agreement. For the two dimensions of innovation activities and innovation capabilities, only the indicator of future use and benefit of these dimensions could be confirmed. In terms of innovation activities, this refers primarily to the acquisition of complementary resources. In terms of innovation capabilities, it is mainly a higher level of knowledge and competence, which SMEs clearly see as beneficial from participation in the Living Lab. It is striking that the indicator regarding the future use and benefit of the acquired innovation capabilities, activities, and knowledge flows is confirmed in all cases. Since this "future use"-indicator measures the respective dimension most directly, the other indicators should be checked for validity due to their high deviation from the "future use"- indicator. In this way, it could be tested whether a different operationalization of the dimensions leads to greater approval and a clearer result.

Due to the small sample of 12 participants and the sampling procedure, the results are limited in their representativeness. Furthermore, no examination of the qualitative criteria could be performed. Besides, many factors from which an effect could be expected from the literature and the qualitative study could not be confirmed in the descriptive analysis of the quantitative study. See the comparison of the results of the qualitative and quantitative study in Table 12. The fact that the effects could not be demonstrated in the quantitative study may also be related to the small sample size. It was too small for the large number of factors examined. Furthermore, due to the different conditions of SMEs and the different

settings and focuses of Living Lab projects, a quantitative analysis of Living Lab projects may not be appropriate.

**Table 12.** Comparison of qualitative and quantitative results.

| Innovation Output | Qualitative | Quantitative |
|---|---|---|
| Improvement of existing product/service/process | | X |
| Introduction of new product/service/process | X | |
| Planned introduction of new product/service/process | | |
| **Innovation activities** | **Qualitative** | **Quantitative** |
| Access to new innovation services | X | |
| Acquire complementary resources | | X |
| Reduce (R&D) costs | | |
| Attract private financing (for R&D) | X | |
| Test product-market fit | | |
| Benefit and use in the future | | X |
| **Innovation capabilities** | **Qualitative** | **Quantitative** |
| Higher level of knowledge and competence | X | X |
| Access to new technologies | | |
| Improved development/innovation process | | |
| Reduced risk of development | X | |
| Create Open Innovation culture | X | |
| Improved attitude towards collaboration | | |
| Benefit and use in the future | | X |
| **Knowledge Flows** | **Qualitative** | **Quantitative** |
| Access to new ideas | X | X |
| Involve users/external viewpoints in innovation process | X | |
| Access to new markets | | |
| Expand network | X | X |
| Exchange knowledge and collaborate | X | X |
| Benefit and use in the future | | X |

From the literature review and the theoretical part of this study, it becomes clear that there is still no uniform definition of Living Labs and that various projects call themselves Living Labs. A definition was created based on characteristics to conduct a target-oriented analysis of the effects on SMEs' innovation performance. This definition concentrates on projects that focus on the method of co-creation, the cooperation of different stakeholders from business, science, and public institutions, and the development in iterations in a real-life setting. It was found that there is little empirical evidence on the effects on the innovation performance of SMEs. Based on the OECD's approach, indicators were developed to measure SMEs' innovation performance resulting from Living Labs. To find out and test the impact of Living Lab projects on them, a qualitative survey of Living Lab experts, i.e., Living Lab managers and participating SMEs, was conducted.

The results of this preliminary study showed that there are various effects of Living Lab projects on the innovation performance of SMEs. Concerning innovation output, the introduction of new products/services or processes was mentioned most often. In terms of innovation activities, the respondents particularly emphasized access to innovation services and testing the product-market fit. In the area of innovation capabilities, the higher level of knowledge and competence and the reduction of risks in the development process were referred to more frequently. Furthermore, more than half of the respondents stated that the Living Lab project had a positive impact on the creation of an Open Innovation culture within their company. The most frequently mentioned effects of Living Lab projects are related to the knowledge flow within the SMEs. Respondents named access to new

ideas, integrating users and external standpoints in the innovation process, expanding the corporate network, knowledge exchange, and collaboration. Nevertheless, the respondents questioned the direct transfer of the method into the work routine. Derived from the qualitative results, the conceptual model based on the program evaluation was refined. The effects were empirically analyzed in a quantitative study.

The structure of the quantitative study only allowed the participation of persons in charge of SMEs. With the quantitative survey's help, some previously established hypotheses could be proven; others had to be rejected or could not be tested. The descriptive analysis of the data showed that Living Lab projects impact the innovation output and knowledge flows of SMEs. It can be deduced that Living Lab projects mainly generate incremental innovation. Among knowledge flows, access to new ideas, expansion of the business network, and knowledge exchange were the most widely agreed upon. The use and exploitation of the generated innovation activities, capabilities, and knowledge flow in the future also received approval. Furthermore, respondents agreed that Living Lab projects positively influence the acquisition of complementary resources and that a higher level of knowledge and competence can be achieved through them. In summary, however, it could not be proven that participation in the Living Lab project has a significant positive effect on SMEs' innovation performance.

With this study, a contribution was made to clarify the research gap of Living Lab projects' impact on SMEs' innovation performance. The development of indicators alone cannot provide information about the potential impact of Living Labs on SMEs' innovation performance which was not available in this way before. Furthermore, it could be empirically proven that Living Labs positively influence innovation output, and knowledge flows within the company. Nevertheless, it could not be confirmed, as hoped by politics and science, that Living Labs have a positive impact on the innovation performance of SMEs that goes beyond the direct results of Living Labs. Furthermore, it was found that a standardized quantitative survey does not provide concise results due to the different settings and priorities within Living Lab projects.

In addition to the new scientific findings, it becomes clear that practical implications can also be derived from the results. This study has discovered that a Living Lab project's funding is not perceived as a reduction of R&D costs for SMEs. This may be related to the high effort of the projects experienced by SMEs, which obscures the benefit of cost reduction. Nevertheless, if desired, policymakers should use the budget in a more targeted way to reduce SMEs' development costs. Thus, they can provide financial support to SMEs that is more visible. The budget can still be used for Living Labs, but it should then also support projects initiated by SMEs with a concrete development focus. The same applies to the issue of sustainability of innovative products or services. An obligatory contribution to the fulfillment of the SDGs would be a good incentive for companies to focus their innovation management even more strongly on the topic of sustainability.

Living Lab managers should focus on providing SMEs with access to the latest technology. This can be achieved by working together with the scientific community or with other companies. The cooperation with science, especially, should be pursued further, to make it clear to the SMEs that this serves as access to knowledge from which they can also profit in the future. Living Lab managers should also dedicate sufficient time to developing a sustainable business model for the innovation to ensure its introduction to the market. Furthermore, Living Lab managers should explain the advantages of the methods used and give them practical tips, e.g., on utilizing the co-creation process in their innovation process in the future. In this way, it can be ensured that SMEs not only receive financial incentives but also, as described in the initial quote, an environment for innovation is created from which they can benefit in the long term.

In order to increase the innovation performance of SMEs in Living Labs, the authors propose a clear definition of objectives, especially with regard to sustainable economic viability beyond the end of the project. This requires a common understanding of all stakeholders regarding the character and intended results of the Living Lab at the beginning

of the project. Above all, the projects should understand themselves as a business model and continuously monitor and iterate it on the basis of predefined innovation indicators (e.g., number of patents, publications, number of external partners, start-ups as well as public relations and co-financing via private-sector partners and the implementation of SDGs). An objective interim and final evaluation by the grant providers would be beneficial in this context in order to give the Living Labs adequate opportunities to react to project failures at an early stage and to adapt the program and process management.

In the future, Living Labs will continue to be an important tool not only for promoting the innovation performance of SMEs but also for helping to shape urban life towards sustainability. Building on the research results of this work, further research topics emerge. The survey of the impact of Living Labs on SMEs' innovation performance could be carried out with a representative sample of Living Lab projects that are not only focused on energy and mobility. A larger sample may allow the identification of correlations among the indicators to clarify the project's priorities. While some Living Lab projects might focus on the innovation output, others might concentrate on teaching SMEs' methodologies to innovate. Furthermore, the results could then be checked for objectivity, validity, and reliability and lead to more meaningful data. The criterion of validity is particularly essential to check for indicators that do not measure what they are supposed to measure. Hence, indicators can be identified that falsify the results, and then a different operationalization of the innovation performance can be considered. Furthermore, an additional weighting of the indicators can be carried out to prioritize them. For instance, the innovation output may have more impact on the overall innovation performance than an expanded network of partners. In addition, an analysis should be carried out at a later point in time to examine to what extent the SMEs have used the impulses generated by the Living Lab project to improve their innovation performance in the long term. It would also be useful to conduct a counterfactual analysis to check whether the increased innovation performance is directly attributable to the Living Lab project or whether another project could have achieved the same result.

**Author Contributions:** Conceptualization, J.A., J.H. and J.K.; methodology, J.A. and J.H.; software, J.H.; validation, J.A., J.H. and J.K.; formal analysis, J.A. and J.H.; investigation, J.A. and J.H.; resources J.A. and J.H.; data curation, J.A. and J.H.; writing—original draft preparation, J.A. and J.H.; writing—review and editing, J.A. and J.H. and J.K.; visualization, J.A. and J.H. and J.K.; supervision, J.A., J.K.; project administration, J.A. All authors have read and agreed to the published version of the manuscript.

**Funding:** This research received no external funding.

**Institutional Review Board Statement:** The study was conducted in accordance with the Declaration of Helsinki. The regulations of the German Science Foundation DFG (https://www.dfg.de/en/research_funding/faq/faq_humanities_social_science/index.html) do not require a statement by an ethics committee for this kind of non-interventional studies.

**Informed Consent Statement:** Informed consent was obtained from all subjects involved in the study.

**Conflicts of Interest:** The authors declare no conflict of interest.

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
