# Peer review of "Living Labs and Small and Medium-Sized Enterprises: A Symbioses Propelling Sustainable Innovation"

_sustainability, doi:10.3390/su141912729_

Round 1

Reviewer 1 Report

The study area is relatively new. This makes development of hypotheses in the first part of the paper a valuable exercise. However, the empirical basis for developing a set of hypoheses has been small.

A more extentive discussion of the different use of Living Labs (LL) in practice is needed prior to analysis, because the outcomes about influence of LLs on SMEs performance may be influenced by different use.

The subsequent testing of the hypotheses is also rather difficult, due to the small number of SMEs participating in the study (n=12). This situation causes an important methodology question: To what extent is the composition of the sample subject to 'self selection bias', influencing validity of the answers?

Another methodological issue is dealing with outcomes of a Likert-scale. Presenting means is not adequate here (as the distance between the points on the scale may be different).

Overall, the abstract is not sufficiently informative (size of samples, method of analysis, country/countries involved, etc.)

The empirical base is too small and may be enlarged by also including medical Living Labs, but this introduces differentation in the outcomes due to the specific sector (as compared to energy and mobility).

Also, the methodology could be extended and improved by using rough-set analysis (fits low level of data and small samples) See e.g. Abbas and Barney, 2016, in Journal of Computer and Communications 4: 10-18.

Further, the analysis of Likert-scale outcomes using means is problematic.

Reviewer 2 Report

The results of the article should been presented in more detail in the Discussion and conclusion section, highlighting possible shortcomings of the research, or future directions of research in more detail. References are recommended to be expanded with more up-to-date sources.
